# The Impact of Point-of-Care Blood C-Reactive Protein Testing on Prescribing Antibiotics in Out-of-Hours Primary Care: A Mixed Methods Evaluation

**DOI:** 10.3390/antibiotics11081008

**Published:** 2022-07-26

**Authors:** Sharon Dixon, Thomas R. Fanshawe, Lazaro Mwandigha, George Edwards, Philip J. Turner, Margaret Glogowska, Marjorie M. Gillespie, Duncan Blair, Gail N. Hayward

**Affiliations:** 1Nuffield Department of Primary Care Health Sciences, University of Oxford, Woodstock Road, Oxford OX2 6GG, UK; thomas.fanshawe@phc.ox.ac.uk (T.R.F.); lazaro.mwandigha@phc.ox.ac.uk (L.M.); george.edwards@phc.ox.ac.uk (G.E.); philip.turner@phc.ox.ac.uk (P.J.T.); margaret.glogowska@phc.ox.ac.uk (M.G.); gail.hayward@phc.ox.ac.uk (G.N.H.); 2Practice Plus Group, Hawker House, 5–6 Napier Court, Napier Road, Reading, Berkshire RG1 8BW, UK; marjorie.gillespie@practiceplusgroup.com; 3Queen Elizabeth Memorial Health Centre, St Michaels Avenue, Tidworth Garrison SP9 7EA, UK; duncangsblair@googlemail.com

**Keywords:** point-of-care tests (POCT), C-reactive protein (CRP), primary care, out-of-hours (OOH), antibiotic stewardship

## Abstract

Improving prescribing antibiotics appropriately for respiratory infections in primary care is an antimicrobial stewardship priority. There is limited evidence to support interventions to reduce prescribing antibiotics in out-of-hours (OOH) primary care. Herein, we report a service innovation where point-of-care C-Reactive Protein (CRP) machines were introduced to three out-of-hours primary care clinical bases in England from August 2018–December 2019, which were compared with four control bases that did not have point-of-care CRP testing. We undertook a mixed-method evaluation, including a comparative interrupted time series analysis to compare monthly antibiotic prescription rates between bases with CRP machines and those without, an analysis of the number of and reasons for the tests performed, and qualitative interviews with clinicians. Antibiotic prescription rates declined during follow-up, but with no clear difference between the two groups of out-of-hours practices. A single base contributed 217 of the 248 CRP tests performed. Clinicians reported that the tests supported decision making and communication about not prescribing antibiotics, where having ‘objective’ numbers were helpful in navigating non-prescribing decisions and highlighted the challenges of training a fluctuant staff group and practical concerns about using the CRP machine. Service improvements to reduce prescribing antibiotics in out-of-hours primary care need to be developed with an understanding of the needs and context of this service.

## 1. Introduction

Anti-microbial resistance (AMR) represents a significant global threat to public health. The global predictive statistical models published in 2022 estimated 4.95 million bacterial AMR-associated deaths in 2019, inclusive of 1.27 million deaths where resistant bacterial infection was the attributable cause, and lower respiratory tract infection was the largest identified contributor to bacterial AMR-associated mortality [1]. Despite efforts to encourage reduced and more appropriate prescribing practices, antibiotics continue to be prescribed in primary care in excess of clinically appropriate levels for self-limiting infections [2], with over-prescribing for respiratory infections being particularly pronounced [3]. Out-of-hours (OOH) services account for 4.5–5.4% of antibiotic prescriptions from primary care in the UK, with higher rates of broad-spectrum antibiotics prescriptions than in-hours care [4], and therefore they represent an important target for implementing strategies to reduce inappropriate prescribing practices [5].

The inflammatory marker CRP has been proposed as a potential tool to help augment clinical assessment in distinguishing between viral and bacterial infection, with levels more markedly elevated in bacterial chest infections, although the evidence is conflicted [6,7,8]. Point-of-care (POC) testing for CRP has been shown in randomised trials to reduce the amount of antibiotics prescribed for lower respiratory tract infections [9,10] and cases of COPD exacerbation in community settings [11].

OOH clinicians have reported a desire for more access to POC testing, including CRP testing to look for infections [12]. However, GPs and OOH staff have also expressed reservations, including concerns about how this supports clinical assessment, alongside worries about test accuracy and interpretation [13,14]. The only study to evaluate POC CRP testing in OOH care focussed on acutely unwell children [15]. We identified one single site’s published report about introducing CRP testing into an OOH primary care base. In this case, the POC CRP tests were intended for use in respiratory tract infections where there was uncertainty about whether the infections were bacterial or viral. They report changes in clinicians’ pre- and post-test prescribing decisions, but do not reporton antibiotic prescription numbers, and they do not include any qualitative data [16].

In collaboration with the Practice Plus Group [17] (an organisation that provides a number of OOH services across England), we implemented a service improvement that offered access to POC CRP tests, which could give results within 3–4 min [18] in three of their OOH assessment bases. We then conducted a mixed methods evaluation of this service improvement.

## 2. Results

### 2.1. Antibiotic Prescribing

Figure 1 and Figure 2 show the number of prescriptions for total antibiotics and respiratory-tract-targeted antibiotics, separately for each base. The fit of the autoregressive integrated moving average (ARIMA) models was satisfactory. Across all bases included in the study, the deviation in the observed prescribing rate in the follow-up period from the mean forecasted trend did not differ between bases with CRP machines and comparator bases. In a before/after comparison of prescribing rates using data from each base individually, there was some evidence of a reduction in prescribing compared to the forecasted trend at two sites; one of these was a base with a CRP machine and the other was a comparator base. Full details of the numbers of prescriptions observed and forecasted are available as Appendix A.

### 2.2. CRP Test Use

In total, 248 tests were recorded on the CRP machines (excluding the test runs used for device training) during the study. The majority of the tests were completed at site A (217/248; 87.5%), whilst 30 (12.1%) were conducted at site B and 1 was conducted (0.4%) at site C. 

Of the 248 tests recorded as completed on the CRP machines, 152 (61.3%) were associated with an entry into the log sheets. Site A documented 124/217 tests (57.1%), site B documented 27/30 (66.7%), and site C documented 1/1 tests. A total of 141/152 (92.7%) logged tests produced usable CRP results. The unusable reported results were due to a non-specific machine-reported error code (in four cases), insufficient sample volumes (2), user errors caused by not turning the machine on (2), problems filling the cartridge (2), and a lack of disinfectant (1). 

Of the 152 tests associated with a data log entry, General Practitioners (GPs) recorded 83 tests (54.6%), Advanced Nurse Practitioners (ANPs) recorded 67 (44.1%), and Emergency Care Practitioners (ECPs) recorded 2 (1%). In total, 23 clinicians conducted tests: 14 GPs, 8 ANPs, and 1 ECP. Over half of all tests (97/152, 63.8%) were conducted by three clinicians working at site A; two of these were GPs who conducted 37 (24.3%) and 16 (10.5%) tests, respectively, and the third, who conducted 44 (28.9%) tests, was an ANP. A full breakdown of which clinicians logged tests is available in Appendix A.

A majority of the tests (112/152, 73.7%) were reported as taking less than 4 min of consultation time. Of the remaining tests, 24 (15.8%) took 4–5 min, 8 (5.3%) took 5–6 min, and 7 (4.6%) took more than 6 min. The data for one test were unclear.

Of the tests, 107/152 (70%) were reported as having been completed in the clinical context of a suspected lower respiratory tract infection. Reasons for the remaining 45/152 (30%) logged tests are given in Table 1.

Where results were obtained, 86/141 (56.6%) of test users documented that the result changed their prescribing decision. This increased to 64% (69/107) when the test was completed for a suspected lower respiratory tract infection. A total of 49 (31.1%) did not change their decision to prescribe after testing, whilst this was unclear for the remaining 6 (3.9%). 

GPs were more likely to report a change in their prescribing decision than other health professionals (i.e., ANP or ECP). GPs changed prescribing decisions 58 out of 77 (75%) times in comparison to 28 out of 64 (44%) times for ANPs or ECPs. Where the test was for a suspected LRTI, the equivalent figures were 46/55 (84%) and 21/46 (46%), respectively.

Of the 248 tests conducted, 59.3% were less than 20 mg/L, whilst 6.5% were over 100 mg/L (see Appendix A).

### 2.3. Qualitative Findings

We conducted 18 interviews with 16 staff, including 12 GPs and 4 allied health professionals (AHPs, including ANPs and ECPs). One GP and one ANP were interviewed twice, towards the beginning and end of the study period. Our sample included staff from all three OOH bases with CRP machines. While all had access to the machines during clinical shifts in the period of the study, our sample included clinicians who told us that they had used the CRP POC equipment often, sometimes, and never. 

### 2.4. The Potential Role(s) for CRP POC Testing in OOH Care

Clinicians identified a potential value for POC CRP testing in OOH care, notably for supporting decision making when there was clinical uncertainty about whether an infection was likely to be viral or bacterial. Although many clinicians were unsure what POC CRP results would add to their clinical assessments, some who had used the testing equipment shared their experiences wherein sometimes there were marked discrepancies between the result and what they had expected the result to be. This could necessitate a review and potential revision of the anticipated pre-test management plan. Mid-range results could be used to support negotiating delayed prescriptions as part of supporting AMS.


*[I]t has made a difference to who I prescribe for, and you know plenty of people who I’d before have said, “Oh go away, you’re absolutely fine,” CRP’s of 75 got standby scripts to take home with them just in case things were getting worse in 48 h And some people I would have prescribed for before just didn’t get any antibiotics...your CRP is less than 5, I know it is viral. Clinician 2 (GP).*


In comparison with in-hours primary care, clinicians explained that the patients they saw were often significantly unwell or had already accessed in-hours care for the same illness episode. At the time when this intervention and evaluation was conducted, patients in OOH care were seen without the benefit of their medical records or previous results, which could increase the potential value of additional diagnostic information.

In addition to supporting clinical decision making, clinicians also explained how test results facilitated conversations about antibiotic prescribing with patients. Having access to an objective or ‘neutral’ result helped them explain why antibiotics were not likely to be helpful, including instances where there was a high expectation of receiving antibiotics from OOH care. 


*The patients seem quite happy with that, once they’ve got something there to look at. I think having the CRP machine is something to back up what you’re saying to the patient. It just helps me to know that I’ve made the right decision as well. Clinician 8 (AHP)*


### 2.5. Considerations about Test Usage

Clinicians, including those who had and had not used the POC CRP machines, expressed concerns about some practical aspects of using the tests. 

An important consideration for many was the time taken to use the machine in the context of the significant pressures of working in OOH primary care.


*I mean given the time constraints, cos you know you can be up to your neck in patients, and you’ve got a waiting room full of patients, and if I’m just faffing around taking blood and taking it there.. I know it doesn’t take very long, but even so it’s still an extra… Clinician 5 (GP)*


A machine’s location impacted time management and how testing could be integrated into the flow of the consultation. At site A, the machine was in a side room. Accessing the machine required clinicians to leave the patient in their room and walk past the receptionist and through the room of patients waiting to be seen.


*You have to leave the patient in the room and [] come out and do the analysis and then I’ve left a patient on their own in my room with my bag there and everything, so practically it’s quite difficult. Maybe if it was actually on the desk it would be more practical than having to leave the patient and go through the waiting room. Clinician 5 (GP)*


At site B, the machine was in one of the clinicians’ rooms for the duration of that shift. These clinicians were better able to integrate performing the test within the flow of the consultation, for example, by examining the patient while the result came through. However, access to testing was effectively limited to the clinician using that room. 


*So, because it’s there [in my room] I tend to use it. If it wasn’t there, I probably wouldn’t walk all the way to the office… when you’ve only got 15 min, it’s two or three minutes more out of your time. Clinician 8 (AHP)*


At site C, the machine was in a side room opposite the coffee room at the end of a corridor and away from the consulting rooms. Only one test was performed at this site.

In addition to issues with the logistics of taking the blood and conducting a test, we heard concerns about how the results fit into clinical care, including reservations about what the results added to clinical diagnostic reasoning. This included concerns about making decisions based on a single CRP value and how best to interpret the CRP results, especially if they were borderline. The lack of specificity of a raised CRP reading was another concern, with clinicians reflecting that the CRP could be raised for many reasons. One GP worried that this might result in more patients being sent to hospital for further assessment. Clinicians were aware that without access to previous results, it was difficult to assess where patients were on their illness trajectory. 

Clinicians also raised concerns about the accuracy of the tests, which reduced their confidence in using them as a basis for clinical decision making, including, specifically, their role in the acute care setting.


*It’s just accuracy of things isn’t it, that you worry about sort of these machines and things….. And you’ve based your decision round that. Clinician 15 (AHP)*


These concerns came together in clinicians’ reflections on the implications for the clinical and medico-legal risk of taking clinical responsibility for POC CRP tests in the OOH care setting.


*[Y]ou did do a CRP and someone comes back and it’s a high CRP, and then goes on to deteriorate, you didn’t send them in, Where do you stand there when you’ve got this, are you more likely to send people in then? I mean you kind of, you’ve got to know what you’re going to do with the results as well, if you start using it as a tool. [] it could go to increasing your uncertainty. Clinician 17 (GP)*


Suggestions that could mitigate some of these concerns included having a machine in every room and having support from a healthcare assistant who could conduct the test. One clinician who had used the machine regularly reflected that the time taken to use the test was often compensated for by reducing time spent on discussions or facilitating onward care. Sharing positive experiences such as when the testing had been helpful was a suggestion for supporting uptake. 

### 2.6. Training Considerations

Although training had been made available, clinicians worried about using the machine (not wanting to risk making a ‘mistake and ruin some expensive equipment’ (GP5)), managing machine supplies and processes, and interpreting the results. We heard about the challenge of offering training and making changes within the OOH care service context, where many clinicians have other roles and undertake irregular or ad hoc shifts. High levels of staff turnover and irregular working patterns complicated both training and the embedding of new processes: 


*[F]ormal training is difficult because lots of people do one session every two weeks in the evening in addition to their day jobs, so training in out-of-hours is tricky. You know to catch everybody you need to run five sessions on something almost. Your chances of finding a time where the ten people who work most regularly are free – not that easy.. Clinician 2 (GP)*


Flexible access to training, for example, recorded sessions which clinicians could access via a remote link, on-site advocacy or mentorship, and regular onsite training were all suggestions for ways to support uptake and confidence. 

In this service innovation, there was no clear protocol or specific guidance about machine usage. Some clinicians felt that this would have increased uptake and could usefully include guidance about a range of clinical situations with evidence for CRP testing. Examples of other areas where it could be helpful to have evidence and guidance included tonsillitis, abdominal pain, and urinary tract infections. Children were identified as a group where CRP results might be helpful, but this would need to be supported by evidence and guidance. In fact, some clinicians explicitly considered these areas where a POC CRP result would be useful, rather than for a possible LRTI. 


*It’s always good to have something in black and white, and I’ll always follow it. It’s good to have guidelines, isn’t it? Clinician 12 (GP)*


Finally, although the offer made to each site was comparable, in terms of equipment, training, and procedures, there was markedly different uptake. Differences in local enthusiasm or project buy-in were identified as elements that influenced uptake in the real-world implementation project. 

## 3. Discussion

### 3.1. Mixed Methods Integration

In this real-world evaluation of a service innovation, we were not able to demonstrate a reduction in antibiotic prescribing in the bases which had access to the CRP testing compared with those that did not. A secondary outcome was to document the number of tests performed. While we do not know the number of consultations conducted during the study period in the three bases with CRP machines in situ, the test numbers represent low uptake, especially in two of the three bases where testing was available. It is likely that insufficient tests were performed to result in a discernible change in antibiotic prescribing rates. While the majority of the tests were completed in the context of respiratory symptoms, our data demonstrate some of the other uses for which clinicians elected to undertake CRP testing when given the autonomy to do so, including sepsis, abdominal pain, and knee pain. 

When tests were completed, predominantly in the context of suspected respiratory tract infection, the majority produced low CRP results, which indicated that there would be no benefit from prescribing antibiotics. The log sheet and qualitative data suggest that for the individuals who did use the testing machine, access to CRP testing could help to change antibiotic prescribing actions. Clinician accounts show how in addition to using CRP results to inform decision making about prescribing antibiotics, they were also used to support negotiating not prescribing antibiotics. Explicit within the NICE suspected LRTI guidance is a CRP range where a ‘delayed’ prescription for antibiotics can be offered as a strategy for reducing antibiotic consumption. Clinician accounts suggest that access to CRP testing could increase confidence in using these ‘delayed’ strategies for antibiotics; however, these would not be identifiable amongst our data and would appear as an antibiotic prescription.

The qualitative data contribute potential explanations for the observed low amount of test usage, including challenges in delivering and maintaining awareness and training within the OOH care context, how time pressures impact decision making about test use, and concerns about the risks implicit in completing a CRP test. There was a suggestion that the site where most tests were completed was developing a more positive culture of testing, with enthusiastic clinicians training others and speaking about the utility of the machine. Alongside culture and training, machine location was a significant factor in test usage and uptake, especially where this influenced time management and clinical pressures in a busy out-of-hours care setting. Having a machine in each consulting room could enable tests to be embedded within the flow of the consultation. Where this is not possible, thinking about the route to the testing kit, or enabling other supports, such as a health care assistant to support the process of testing, could be worth considering. 

While we are unable to determine causality, our observational data show that site A, which had the highest test usage, also had the largest reduction in antibiotic prescribing. The drop in observed antibiotic prescriptions exceeded the number of tests completed, and so it cannot represent a straightforward association, but it could reflect a trigger towards a shift in practice. 

### 3.2. Strengths and Limitations

The strengths of this study include that it is a real-world evaluation of an antimicrobial stewardship technology embedded into an OOH primary care setting. The mixed methods approach allowed us to explore multiple impacts of POC CRP testing. We spoke to a range of staff, including some at both the beginning of the project and at the end. The antibiotic data represent a complete ascertainment of antibiotic prescribing during the study period. 

The study’s limitations include that as it was observational in nature, we cannot causally attribute changes in prescribing to the use of CRP testing. Although we attempted to mitigate this by using a comparative interrupted time series design, there may be underlying location-specific trends in prescribing that our analysis could not account for. For example, as we were unable to determine the number of consultations that the number of antibiotic prescriptions were related to, we could not control for changes in consultation rates during the follow-up period. The low number of tests performed limited our ability to detect an impact on the antibiotic prescribing rates. While clinicians reported whether the result changed their prescribing decision, we do not have consistent data for how or in what way their decision may have changed, including whether this reflected changes in their communication approach or navigating different strategies within prescribing. Whilst we aimed for variation in our qualitative interviewees, we recruited a small number of clinicians, which may not have been representative of all clinicians who had access to POC testing. 

### 3.3. Comparison with Existing Literature

POC CRP tests have been shown to reduce antibiotic prescribing in primary care, especially alongside educational interventions [11]. CRP testing in primary care settings alongside clinical guidance is associated with a reduction in immediate antibiotic prescribing for acute respiratory infections, in both adults and children, without any apparent increase in patient reported adverse effects or re-consultation rates, though a small increase in hospital admission rates could not be ruled out [19,20]. A recent meta-analysis supports the conclusion that the benefits of using POC CRP testing in suspected respiratory tract infections outweigh the potential harms, although reduced antibiotic prescribing balanced against an increase in re-consultation rate. This analysis did not find an increase in rates of hospitalisation [21]. POC CRP testing is cost effective in the primary care management of respiratory tract infections [22]. In the OOH setting, a UK case report documents the introduction of a POC CRP machine into an OOH base, accompanied by guidance about the CRP prescribing thresholds described in NICE CG 191, as in our study. This project was trialled in the context of an increase in antibiotic prescribing in OOH care (especially broad-spectrum antibiotic prescribing) in their locality at a time when in-hours prescribing was falling. This study reports a tendency towards changing the pre-test prescribing decision from prescribe to not prescribe, albeit with some reversals in the opposite direction. They suggest that costs (and test use) were lower than anticipated. The authors postulated that a possible explanation for this is that cases of true clinical equipoise after examination and history are the exception [16]. A study in Northern England in-hours care found that test use uptake was variable, concluding that the small sample size due to non-test use made assessing impacts on the effect of the test availability on antibiotic prescribing hard to evaluate. They concluded that staff would need greater support and resources [23].

OOH primary care is an important area of service delivery in which to explore effective strategies for antimicrobial stewardship because of its high rates of antibiotic prescribing. It is important to reflect that this may not represent a greater proportion of ‘inappropriate’ prescribing because the differences in case mixes between in- and out-of-hours care mean that a straightforward comparison may not be appropriate. This has been found to be the case in the Netherlands, where an analysis of national prescribing data did not show a difference in the quality of prescribing, but that rates of antibiotic issue reflected differences between the presenting population of patients [24].

A previous assessment suggested a decrease in OOH antibiotic prescribing rates in England between 2010 and 2014 [4], although a separate evaluation in Oxfordshire indicated local variability, was possibly caused by a displacement in prescribing from in-hours care [25]. In our study, while many of the OOH bases exhibited a reduction in prescribing rates between 2009 and 2019, there was high variability between the sites. This highlights the importance of multi-site evaluation in studies of this type, and the factors that underpin this variation remain understudied.

The clinicians in our study described how OOH primary care can be experienced as a high risk or challenging clinical environment in which to work. This resonates with previous accounts of clinical decision making in the OOH context, where priorities include ‘fire-fighting’ and needing to determine how acutely unwell someone is and rule serious illness in or out [26]. The clinicians we spoke to reflected on how consultations in OOH primary care are complicated by not having access to patients’ medical records and the inability to follow patients.. This aligns with other qualitative work exploring primary care clinician perspectives in OOH care [27].

Surveys conducted with clinicians, including OOH clinicians, suggest that they want to have greater access to POC tests [12,28]. However, a previous real-world evaluation of a service innovation introducing POC tests (not including CRP) into an English OOH setting also found low rates of test use and comparable concerns from clinicians about the reliability of test results, the medico-legal risk, and what they add to clinical diagnostic reasoning. As with our present study, there was a developing understanding of how and when the tests added value. Harnessing these reflections could be used to promote uptake and develop a positive culture surrounding test use [14,29]. 

In our interviews, test results had value as both an adjunct to diagnosis and an aidin communicating with patients where they supported navigating shared decision making by offering ‘objective’ test results. This aligns with previous work about CRP testing in primary care where the results could help GPs when they were promoting the advice that antibiotics were not needed [30,31]. 

Embedding AMS practices through education requires the consideration of both formal learning and mentorship and role-modelling [32]. These are both likely to be relevant to the OOH and urgent care settings, as highlighted within our clinician interviews. There is a documented lack of specific guidance for OOH staff about antibiotic prescribing, as well as a need to understand the key features required for OOH context-specific training and resources [33]. In a qualitative study with OOH clinicians about their experiences in prescribing antibiotics for respiratory tract infection symptoms, Williams et al. identified factors specific to the OOH care service. These included the lack of records, lack of follow-up or feedback systems, and a mixed workforce, leading the authors to conclude that interventions for reducing antibiotic prescribing would need to be tailored to meet the needs of this care setting [27], which resonates with our study findings. The findings from this study will likely have resonance with other settings undertaking primary care and urgent assessments. For example, ‘unscheduled’ primary care may be delivered in hubs or by extended primary care teams both during standard and extended practice hours, as well as in OOH care settings.

### 3.4. Implications for Research and Practice

OOH and urgent primary care settings merit a greater focus on the development and evaluation of antibiotic stewardship approaches. Future work to evaluate AMS interventions should take into account the variability in antibiotic prescribing trends we have demonstrated herein and include sufficient sites to ensure confidence in any observed reductions in prescribing being linked to the new intervention. In this study, we observed a wide range of test usage across three sites.

Our interview data suggest that future research and practice might encourage more widespread adoption if machines are sufficient in number to allow each clinician to have one on their desk or if they are embedded so that they do not add time or complexity to the processes of consultations. This should also include consideration of the impacts of evolving approaches to primary care delivery following the COVID-19 pandemic, such as the move towards more remote forms of consultation.

Our interviews suggested that some clinicians may have preferred more structured guidance regarding POC CRP, including how to interpret and utilise the results, and the potential application of POC CRP testing to a much wider range of conditions than just LRTI. A worthwhile focus for future research would be the development and evaluation of bespoke OOH care and urgent primary care guidance and training on antibiotic stewardship, including the use of diagnostic tools and clinical mentorship throughout the OOH team. Research is needed to understand what resources and evidence clinicians would find helpful for navigating the balance between clinical discretion and evidence-based guidance and how these could usefully be shared and disseminated.

## 4. Conclusions

We were unable to confidently demonstrate that introducing POC CRP testing into primary care OOH centres was associated with a reduction in antibiotic prescribing, in part because of the relatively low uptake and test usage. However, when tests were completed, they were found to be valuable in managing both diagnostic equipoise and navigating communication around not prescribing antibiotics. Many of the barriers to test utilisation identified in this study are amenable to mitigation by services considering the implementation of POC CRP testing. Future actions could include flexible training and education, machine location, mentorship, and sharing positive experiences of test use.

## 5. Materials and Methods

Eight primary care OOH bases under the governance of the Practice Plus Group were selected a priori for inclusion in the study on the basis that they had similar monthly antibiotic prescribing rates in adults (since prescribing data became available in 2009). Three of these bases were allocated an Afinion™ 2 point of care analyser with Afinion™ CRP cartridges (Abbott Diagnostics Technologies AS, Oslo, Norway) between September 2018 and December 2019. These three bases were selected because they were in the same administrative region in order to facilitate implementation. The other five bases did not receive a CRP machine and acted as comparators. Service provision at one of the comparator bases was discontinued during the follow-up period, and so this base was subsequently excluded. Further details about the selection of bases for inclusion in the study are provided by Fanshawe et al (2022) [34]. 

At bases with CRP machines, clinicians were trained in how to use the machine and provided with a summary of NICE suspected pneumonia guidance (NG 191), which outlined how to use thresholds of CRP to determine the need for antibiotics in LRTI [19], but clinicians had free choice about when and whether to use the machines. Clinicians were advised that the administration supported the use of POC CRP testing in adults and children where clinicians felt it would enhance their decision making or inform their care management.

We undertook a mixed methods evaluation of this service improvement project using a concurrent design whereby the quantitative and qualitative data were collected in parallel and synchronously throughout the same study period. They were initially analysed independently from each other prior to the mixed methods integration. 

## 6. Outcomes

### 6.1. Quantitative

The primary outcomes were the number of respiratory-tract-targeted antibiotic prescriptions and the total number of antibiotic prescriptions issued in adults, which were measured relative to the expected number based on the time trend at each OOH base and by comparing bases with CRP machines to bases without machines. Prescription data are routinely recorded and therefore reflect complete data on antibiotic prescriptions given to adults at participating bases during the study period. The secondary outcomes included the number of CRP tests performed and machine failure rates. 

### 6.2. Quantitative Data Collection and Analysis

We obtained the numbers of antibiotics prescribed monthly at each included base. Respiratory-tract-targeted antibiotic prescriptions were those prescriptions for the antibiotics listed in Appendix A. We adapted methods for comparative interrupted time series to model the time series of monthly antibiotic prescriptions at each base until August 2018 using an autoregressive integrated moving average (ARIMA) model, allowing for seasonality. From this, we obtained the predicted number of antibiotic prescriptions and its standard error for each site during the prospective phase of the study, against which the observed number of prescriptions during this period was compared using Z-tests for each base individually and for the overall comparison of bases with CRP machines versus those without. Analyses were conducted in R [35] using the R packages ‘forecast’ [36] for model fitting and ‘ggplot2’ [37] for generating plots. For full details of the analytical approach adopted, see Fanshawe et al. (2022) [34].

The number of tests completed was recorded within the three machines in situ at the Care UK OOH bases. 

Clinicians were requested to complete a brief log sheet when they performed a CRP test, which documented the following variables: date and time of test, time taken out of consultation for the test (in minutes), professional status of test user, whether the prescribing decision was changed as a result of the test result, and whether the test was for a suspected lower respiratory tract infection (or if not, to specify the reason).

### 6.3. Qualitative Data Collection and Analysis

We conducted semi-structured interviews with clinicians working at the three bases that were provided with a CRP machine. We spoke to GPs, advanced nurse practitioners (ANPs), and extended care practitioners (ECPs). Interviews were offered either face-to-face, where they were conducted on-site at the OOH base, or over the telephone. The interviews used a semi-structured topic guide (included in the Appendix A) developed from our clinical and research experience. The interviews were conducted by an SD (GP and qualitative researcher). The site visits and associated face-to-face interviews were conducted between 18 January and 13 December 2019. The participants provided their consent for participation, interview transcription, and the use of quotations in publications. 

The interviews were audio-recorded and transcribed verbatim with the participants’ consent. A coding framework was iteratively developed in NVivo12 by the SD/MG based on expected and emergent themes. The data were analysed thematically [38] using mind-mapping techniques [39], and the qualitative findings were reviewed by the study team, which included GPs, data scientists, and social science researchers. 

### 6.4. Mixed Method Integration

Following data collection and an initial independent analysis of the quantitative and qualitative data analysis, the study team met, shared their findings, and considered how these might inter-relate and be utilised to postulate potential explanatory hypotheses or further research questions. We considered areas of convergence where the results offered potential explanations for phenomena seen in both datasets and areas of divergence, including the implication of these for the development of further hypotheses. 

The qualitative arm of the study received ethical approval from the University of Oxford Medical Sciences Interdivisional Research Ethics Committee: CUREC R55414/RE002. The quantitative data collection and analysis were prospectively approved as a service evaluation by the Practice Plus Group. 

## Figures and Tables

**Figure 1 antibiotics-11-01008-f001:**
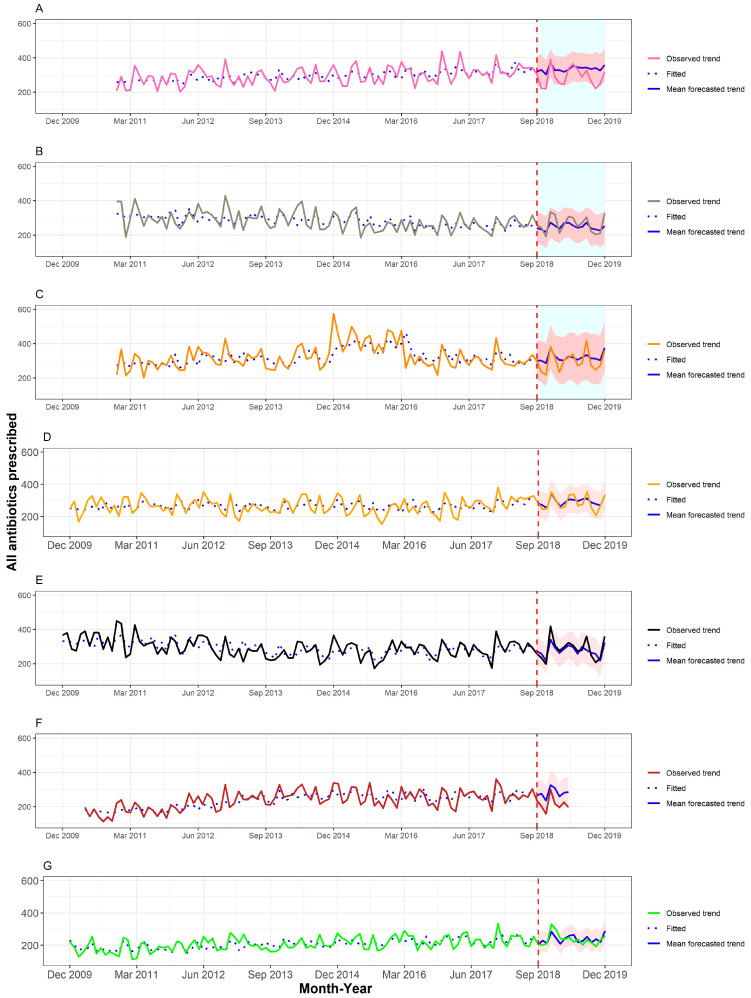
Time series for all antibiotics prescribed for adult patients across the different OOH primary care locations with CRP machines, (**A**–**C**), and those without CRP machines, (**D**–**G**). The fitted values obtained from the ARIMA models are depicted using a dotted blue line (pre-September 2018), and the forecasted mean trends are depicted using a solid blue line (post-September 2018), with the corresponding 95% confidence interval highlighted in red.

**Figure 2 antibiotics-11-01008-f002:**
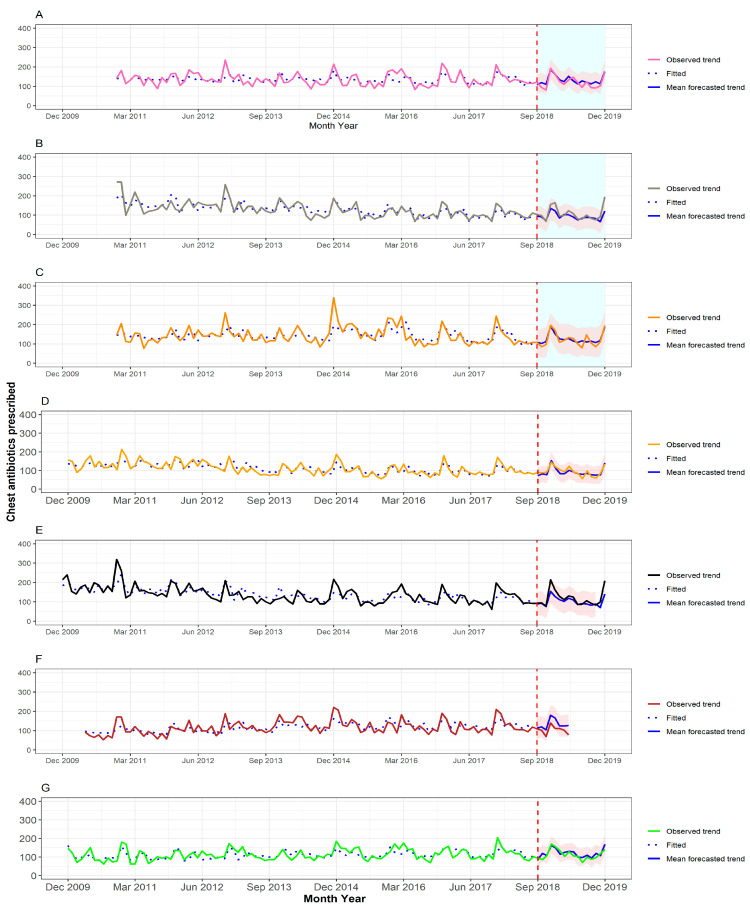
Time series for respiratory-tract-targeted antibiotics prescribed for adult patients across the different OOH primary care locations with CRP machines, (**A**–**C**), and those without CRP machines, (**D**–**G**). The fitted values obtained from the ARIMA models are depicted using a dotted blue line (pre-September 2018), and the forecasted mean trends are depicted using a solid blue line (post-September 2018), with the corresponding 95% confidence interval highlighted in red.

**Table 1 antibiotics-11-01008-t001:** Log sheet of the recorded reasons for doing a POC CRP test.

Reason	N	%
Lower respiratory tract infection	108	71%
Not reported	10	7%
Cough	5	3%
Abdominal symptoms	5	3%
Upper respiratory tract infection	4	3%
Reassurance or advice	4	3%
Sinusitis	2	1%
Temporal arteritis	2	1%
Tonsillitis	2	1%
Confusion	1	1%
Cystic Fibrosis	1	1%
Diverticulitis	1	1%
Knee pain post-operation	1	1%
Meningitis	1	1%
Recurrent ear pain/headache	1	1%
Sepsis	1	1%
Urinary tract infection	1	1%
Uvulitis	1	1%
Vasculitis	1	1%

## Data Availability

Third Party Data: Restrictions apply to the availability of these data. The data were obtained from the Practice Plus Group and are available from the authors with the permission of the relevant stakeholders. The qualitative data are not publicly available due to reasons of privacy and in line with ethical approvals. The corresponding author may be contacted with any queries.

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
