# Peer review of "The Impact of Point-of-Care Blood C-Reactive Protein Testing on Prescribing Antibiotics in Out-of-Hours Primary Care: A Mixed Methods Evaluation"

_antibiotics, 2022, doi:10.3390/antibiotics11081008_

Round 1

Reviewer 1 Report

On the manuscript
1. Line 507. the word’’ References must be numbered in order of appearance in the text (including citations in tables and legends) and listed’’ need to be removed and ref no.1 need to be started on the next line separated from ref guideline mentioned
2. Reference numbering need to be rechecked as only one numbering is needed
3. Line 391. Should materials and methods been placed after the results and discussion sessions? Better to be read before results and discussion for better understanding of study procedure and other essential information
On the supplementary document
1. On supplementary table 2, location data on far-left column need to be in an alphabetical order or need to be explained the reason for not being in order?
2. Table 1 to 3 need a clarification whether they were missing or labeling wrongly. I could not find these tables on the document.
3. On supplementary table 3, location data on far-left column need to be in an alphabetical order or need to be explained the reason for not being in order?

Author Response

Review 1

Comments and Suggestions for Authors

On the manuscript
 1. Line 507. the word’’ References must be numbered in order of appearance in the text (including citations in tables and legends) and listed’’ need to be removed and ref no.1 need to be started on the next line separated from ref guideline mentioned

Thank you for your observation and suggested corrective action, we have amended the references section as requested.

  1. Reference numbering need to be rechecked as only one numbering is needed

This has been updated, thank you.

  1. Line 391. Should materials and methods been placed after the results and discussion sessions? Better to be read before results and discussion for better understanding of study procedure and other essential information

We have cross-checked against the journal requested order of sections, and can confirm that our order with methods after the discussion, is consistent with the journal style and other recently published articles.

 On the supplementary document
 1. On supplementary table 2, location data on far-left column need to be in an alphabetical order or need to be explained the reason for not being in order?

We have changed the order in the table to alphabetic order as suggested by the reviewer.

  1. Table 1 to 3 need a clarification whether they were missing or labeling wrongly. I could not find these tables on the document.

Thank you. We have amended the files and feel that we have resolved this question, however please let us know if any confusion remains.

  1. On supplementary table 3, location data on far-left column need to be in an alphabetical order or need to be explained the reason for not being in order?

We have changed the order in the table to alphabetical order as suggested by the reviewer.

Reviewer 2 Report

I read with interest the paper. I find it well wrote with interesting data and experience 

Below my minor suggestions

1. Introdutcion: if you can show some data on global AMR burden 

2. Methods and results: clear

3. Discussion: discuss better the difference between colonization and infection and the crucial role of infection prevention control to prevent and control antimicrobial resistance. Also underline the role of training on this important global health problem

4. Limitations section: well done

5. Conclusion: give some proposal that came from your interesting paper, and stress if you agree the role of medical education on AMR as key to control AMR spread especially in young generation of doctors and nurse (see and cite if you want Italian young doctors' knowledge, attitudes and practices on antibiotic use and resistance: A national cross-sectional survey. J Glob Antimicrob Resist. 2020 Dec;23:167-173)

Author Response

Review 2

Comments and Suggestions for Authors

I read with interest the paper. I find it well wrote with interesting data and experience

Below my minor suggestions

  1. Introduction: if you can show some data on global AMR burden

We thank the reviewer for suggesting this important addition to the paper. We have added the following passage to the introduction on page 1 of the manuscript to better illustrate the global burden of AMR:

“Global predictive statistical models published in 2022 estimated 4.95 million bacterial AMR associated deaths in 2019, inclusive of 1.27 million deaths where resistant bacterial infection was the attributable cause; lower respiratory tract infection was the largest identified contributor to bacterial AMR associated mortality” – reference, Murray, C. J. et al. Global burden of bacterial antimicrobial resistance in 2019: a systematic analysis. Lancet 399, 629–655 (2022).

  1. Methods and results: clear
  2. Discussion: discuss better the difference between colonization and infection and the crucial role of infection prevention control to prevent and control antimicrobial resistance. Also underline the role of training on this important global health problem.

Thank you for this suggestion. This study evaluated CRP which is a non-specific marker of host inflammatory response and therefore we believe, although important, that the debate about colonisation versus infection is not relevant to this paper. Similarly, infection prevention and control, whilst important, is not a key feature of these types of acute community services.

We agree that training is important and have extended our comment regarding the lack of high-quality evidence in this area. Thank you for your suggestion of a reference that highlights the importance of considering both formal education and role modelling in AMS stratagems in clinical practice, which we have included in this revised version of our paper. Please see this sentence added to the comparison with other literature section:

“Embedding AMS practices through education needs consideration of both formal learning but also mentorship and role-modelling. These are both likely to be relevant to the OOH and urgent care setting, as highlighted within our clinician interviews.” (page 10)

We have followed this reflection through into the implications for further practice section as below:

“Our interviews suggested that education is likely to be central to the successful implementation of AMS strategies, and s some clinicians may have preferred more structured guidance regarding POC CRP, including how to interpret the results and use them, and the potential application of POC CRP testing to a much wider range of conditions than just LRTI. A worthwhile focus for future research would be the development and evaluation of bespoke out of hours and urgent primary care guidance and training on antibiotic stewardship, including the use of diagnostic tools and clinical mentorship within the roles. Research is needed to understand what resources and evidence clinicians would find helpful for navigating the balance between clinical discretion and evidence-based guidance and how these could usefully be shared and disseminated.” (page 11)

  1. Limitations section: well done
  2. Conclusion: give some proposal that came from your interesting paper, and stress if you agree the role of medical education on AMR as key to control AMR spread especially in young generation of doctors and nurse (see and cite if you want Italian young doctors' knowledge, attitudes and practices on antibiotic use and resistance: A national cross-sectional survey. J Glob Antimicrob Resist. 2020 Dec;23:167-173)

Thank you for this guidance. We agree that education has a critical part to play in AMS, and that strategies will benefit from both initial investment and ongoing support in learning and service delivery. We have amended the closing paragraph of the implications for practice section to reflect this:

“Our interviews suggested some clinicians may have preferred more structured guidance regarding POC CRP, including how to interpret the results and use them, and the potential application of POC CRP testing to a much wider range of conditions than just LRTI. A worthwhile focus for future research would be the development and evaluation of bespoke out of hours and urgent primary care guidance and training on antibiotic stewardship, including the use of diagnostic tools and clinical mentorship within the roles. Research is needed to understand what resources and evidence clinicians would find helpful for navigating the balance between clinical discretion and evidence-based guidance and how these could usefully be shared and disseminated.” (page 11)

In addition, we have added a conclusion section, which reinforces this message:

“Conclusion

We were unable to confidently demonstrate that introducing POC CRP testing into primary care out of hours centres was associated with a reduction in antibiotic prescribing in this evaluation, in part because of the relatively low test usage. However, when tests were done, they were found to be valuable in managing both diagnostic equipoise and in navigating communication around not prescribing antibiotics.  Many of the barriers to test utilisation identified in this study are amenable to mitigation by services considering implementing POC CRP. Actions could include flexible training and education, machine location, mentorship, and sharing positive experiences of use”. (page 11)